# Process to Establish the Enhance of Fatigue Life of New Mechanical System Such as a Drawer by Accelerated Tests

Seongwoo Woo [1,*], Dennis L. O'Neal [2] and Yimer Mohammed Hassen [1]

1   Manufacturing Technology, Mechanical Technology Faculty, Ethiopian Technical University,
    Addis Ababa P.O. Box 190310, Ethiopia; yimoha@ymail.com
2   Department of Mechanical Engineering, School of Engineering and Computer Science, Baylor University,
    Waco, TX 76798-7356, USA; dennis_oneal@baylor.edu
*   Correspondence: twinwoo@yahoo.com; Tel.: +251-90-047-6711

**Abstract:** To extend the life of a mechanical system, parametric Accelerated Life Testing (ALT) is proposed as a procedure to identify design faults and reduce fatigue failure. It includes a derivation of generalized time to failure model by linear transport process and a sample size equation for the ALT. A refrigerator drawer was used as an example. After loading, the rail rollers broke and the center support was bent. Ribs were added to the center support and the rail roller support was extended. At the first ALT, the box cover failed near the intersection between the cover and body. The box was then modified by increasing the rib and fillet. At the second ALT, the rails and center support fractured. They were altered by increasing the rib and corner rounding. After the third ALT, there were no issues. The drawer lifetime was ensured to be B1 life 10 years.

**Keywords:** fatigue; parametric ALT; mechanical system; drawer; design faults

## 1. Introduction

Because of increasing competition in the market, new features in refrigerators are often quickly brought to market for the end-users. With inadequate testing or no comprehension of how the new features are utilized by the consumer, the new features can grow product failures in the marketplace and negatively affect the manufacturer's brand name. These new attributes for the product might be evaluated in the design phase before being sold into the market. Systematic methods with reliability quantitative (RQ) specifications for utilizing an established mechanical system need to be included in evaluating new features.

Mechanical systems, such as automobiles, airplanes, and refrigerators [1], transfer power to achieve a desired outcome. Forces are used to provide movement of mechanisms that are typically subjected to repeated loads. In the process, fatigue failure may occur if there is a stress raiser such as notched, sharp-edged, or thin surface, etc., in a component. Mechanical systems are made of multi-module structural systems. While these systems can be complex, a mechanical system can perform satisfactorily if it is designed appropriately. For instance, by making use of the vapor-compression refrigeration cycle, a domestic refrigerator can provide cooling and freezing for the food stored in the refrigerator. The evaporator provides cold air to both the refrigerator and freezer sections. A refrigerator includes several subsystems—door, cabinet, drawers and shelves, control system, compressor or electric motor, heat exchanger, water supplying device, and other components.

A refrigerator might be composed of as many as 2000 components. Product life is targeted to be no less than B20 life 10 years. A refrigerator is made up of 20 units (or 8~10 modules) with each unit having approximately 100 parts (See Figure 1a). Therefore, the lifetime target of each unit should have B1 life 10 years because the system life is controlled by any design defects in any module. Figure 1b shows a newly designed module #3, which has a design defect and the shortest time before failure. As a result, this module determines the system lifetime for the whole refrigerator.

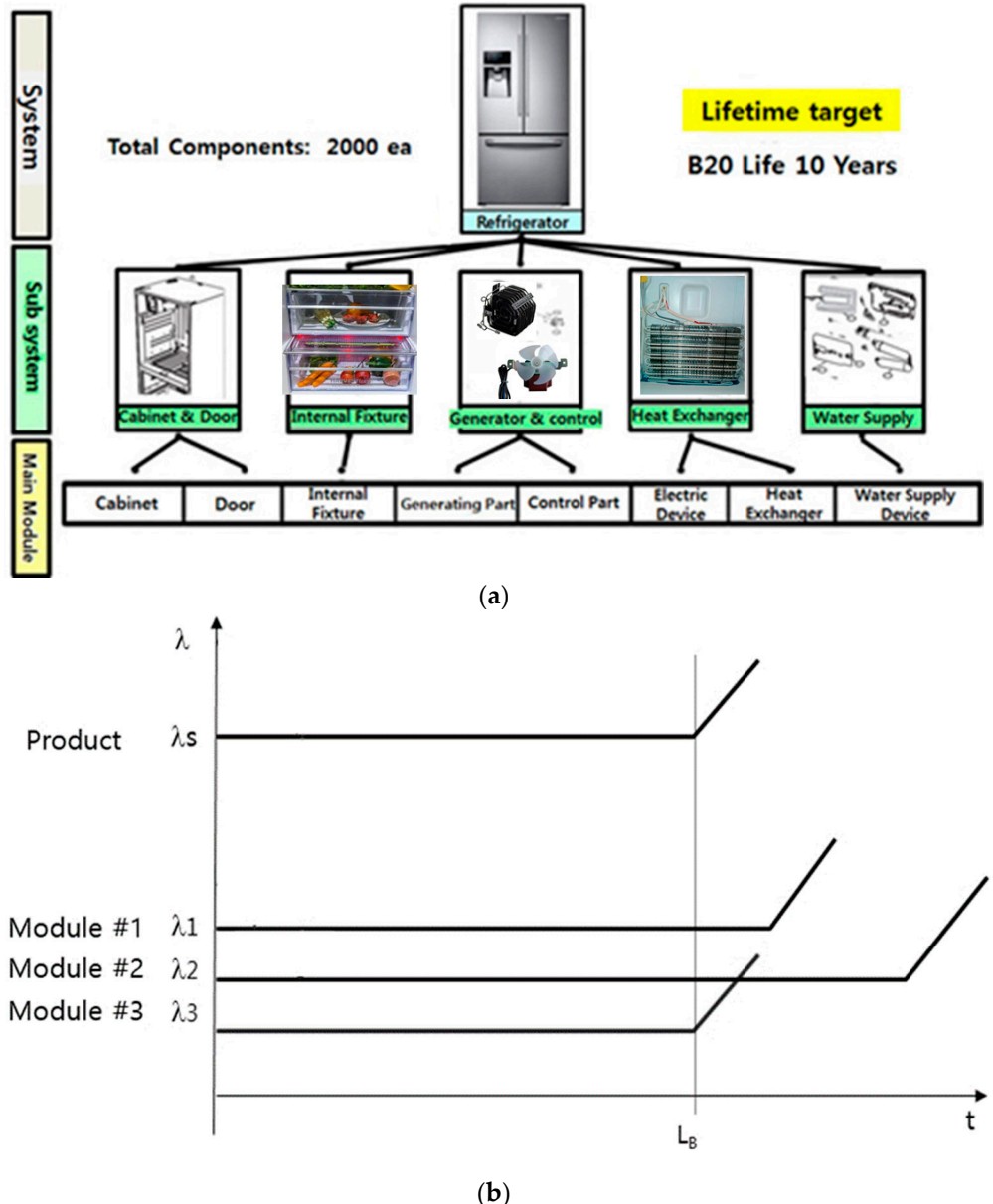

**Figure 1.** Product lifetime with multi-modules resolved by new module (**a**) grouping of multi-module refrigerator; (**b**) product life $L_B$ and failure rate $\lambda_s$.

To prevent recalls of a mechanical product from the market [2], the modules comprising the product should be designed to survive typical operations by the end-users who acquire and make use of the product. If the product has design defects that will cause failure, it should be verified by proper parametric accelerated life testing (ALT) [3,4]. Fatigue is often the principal origin of failure in metallic components, accounting for approximately 80–95% of all structural failures [5]. It manifests itself in the shape of cracks which originate from high stress concentrations such as grooves, thin surfaces, holes, etc. in mechanical systems and propagate it to the end. Fatigue thus is the weakening of a material caused by cyclic loading.

A great deal of attention is currently being given to the performance of low-cycle fatigue of super-alloys, particularly in the field of turbine–engine designs made of nickel-base polycrystalline [6–9]. The failure criteria are often evaluated by the representative Goodman diagram [10,11]. However, it is difficult to estimate the lifetime cycles of multi-module products because small samples of the part, not the module, are tested, and part

failures due to design defects rarely occur in the marketplace. These methodologies often fail to reproduce the market failures because comprehensive testing might have been necessitated, which might be too expensive due to limitations of time and sample size.

Fatigue failures also depend on measurable factors such as the cyclic stress amplitude, mean stress, or stress ratio, R (= $\sigma_{\min}/\sigma_{\max}$), which might be expressed as the ratio of the minimum cyclic stress to the maximum cyclic stress [12]. Utilizing an elevated load ratio which may be expressed as an accelerated factor (AF) may help identify the design flaws such as stress raisers in the structural component.

Any design defects might be identified and altered by the Taguchi approach [13] or design of experiments (DOE) [14] before a commercially manufactured good is delivered. However, these approaches demand large computations for optimal solutions. However, they might not be useful because they may not identify an underlying failure mechanism such as fatigue. If there are design defects which create a lack of stiffness (or strength) when a system is subjected to repeat loading, the system will be unsuccessful before its expected life due to fatigue.

Engineers have often used traditional design aspects such as strength of materials [15,16]. A recent fracture mechanics investigation [17,18] showed that the key components might be fracture toughness as an alternative of strength as an applicable material property. With the use of quantum mechanics in the electronic industry, designers have identified that product failures occur from micro-void coalescence (MVC). This type of failure is also observed in a great number of engineering plastics or metallic alloys. To better identify the failures in a mechanical system, a life-stress model [19] might be integrated with a conventional design perspective and connected to failure of electronic elements due to a pre-existing defect or little crack. This process would not be practical for use in a finite element method (FEM) approach [20,21].

To evaluate product failures, there is another engineering viewpoint which can be used along with the FEM. Several engineers have proposed that rare system failures might be assessed by: (1) rigorous modeling such as Newtonian or Lagrangian techniques; (2) getting its time response for (dynamic) loads, acquiring the system stress/strain from it; (3) making use of the rain-flow counting procedure for von-Mises stress [22,23]; (4) estimating product damage by the Palmgren–Miner's rule [24]. Nevertheless, using a methodology which may provide closed-form, accurate solutions would require applying several assumptions which might not recognize multi-module system failures, due to material defects such as contacts, thin surface, micro-voids, etc., as subjected to repeated loading.

This research presents parametric ALT as a systematic approach which may be used to recognize design faults of a newly designed mechanical structure and modify them. It includes: (1) an ALT scheme created on system BX lifetime, (2) load check for the ALT, (3) suitable modified ALTs with the revised designs, and (4) an appraisal of whether the newest design(s) of the system fulfils the targeted BX lifetime. We also suggested generalized time to failure model and sample size formulation. As a test investigation, a newly designed drawer system in a refrigerator subjected to repeated food loading is evaluated.

## 2. Parametric Accelerated Life Testing for Mechanical Product

### 2.1. Meaning of BX Lifetime

To implement parametric ALT, BX life as a measure of system life is necessitated. BX life, $L_B$, may be viewed as the amount of time when X percent of a collection of a selected product might have failed. Otherwise, 'BX life Y years' is a preferable term for system life, which helps to satisfactorily decide the cumulative failure rate of a product in response to its use in the field. As an example, if the product lifetime has B20 life 10 years, 20% of the population might have been unsuccessful in achieving one's goal for 10 years of a working period. On the other hand, the mean time to failure (MTTF)—B60 lifetime—as the reverse of the failure rate might not be utilized for the system lifetime. It is too lengthy a time to be

unsuccessful in the 60% of the population in commercially manufactured products. BX life should indicate a more proper measure for product lifetime, contrasted with MTTF.

### 2.2. Placing an Entire Parametric ALT Scheme

Reliability might be described as the capability of a system to function under specified circumstances for a stated interval of time [25]. It has traditionally been illustrated by the "bathtub curve" (Figure 2) which has three sections [26]. In the first section, there is a decreasing failure rate in the earlier portion of the product's life ($\beta < 1$). In the second section, there is a flat failure rate ($\beta = 1$) during the middle life of the product which follows an exponential distribution. Eventually, there is an increasing failure rate to the termination of the product's life ($\beta > 1$) which follows a Weibull distribution.

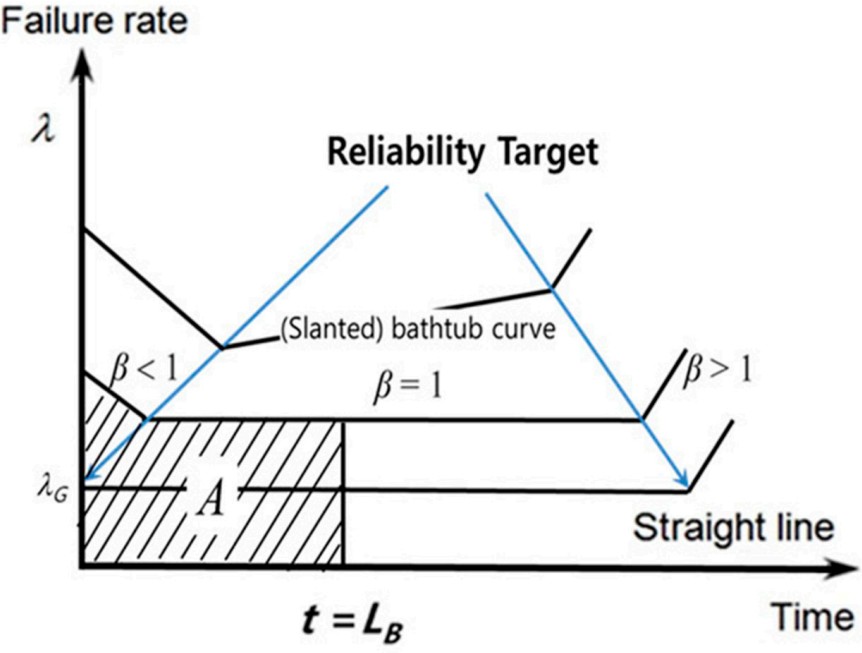

**Figure 2.** Reliability Index; BX Life ($L_B$) on the bathtub.

Assuming that $T$ is a random variable specifying the time to failure, the reliability function, the fraction yet surviving at time $t$, could be expressed as follows:

$$R(t) = P(T > t) \tag{1}$$

As the complement of $R(t)$, the accumulative distribution function (CDF), $F(f)$, might also be defined as:

$$F(t) = 1 - R(t) \tag{2}$$

For individuals in a population has a density function $f(t)$. The equivalent distribution function is the fraction of the population failing by the time $t$. That is,

$$F(t) = \int_{-\infty}^{t} f(s)ds \tag{3}$$

The failure rate function $\lambda(t)$ measures the instantaneous risk, in that $\lambda(t)\delta t$ is the chance of failing in the following small interval $\delta t$ given survival to time $t$. From the relation we can find as follows:

$$P(\text{survival to } t + \delta t) = P(\text{survival to } t)\, P(\text{survival to } \delta t \mid \text{survival to } t) \tag{4}$$

where $P$ is probability

So we can obtain the following equations. That is,

$$1 - F(t + \delta t) = \{1 - F(t)\}\{1 - \lambda(t)\delta(t)\} \tag{5a}$$

$$\delta t F'(t) = \{1 - F(t)\}\lambda(t)\delta(t) \tag{5b}$$

So, the failure rate function is expressed as follows:

$$\lambda(t) = \delta(t)F'(t)/\{1 - F(t)\}\delta(t) = f(t)/\{1 - F(t)\} = f(t)/R(t) \tag{6a}$$

$$\text{or } \lambda(t) = \frac{f(t)}{R(t)} = \frac{dF(t)/dt}{R(t)} = \frac{(1 - R(t))'}{R(t)} = \frac{-R'(t)}{R(t)} \tag{6b}$$

If Equation (6b) is integrated with respect to time, the *X*% accumulative failure $F(L_B)$ at *BX* life, $L_B$ may be estimated.

$$F = \int \lambda(t)dt = -lnR(t) \tag{7a}$$

In other words, we might state $F(L_B)$ as:

$$A(orX) = \langle \lambda \rangle \cdot L_B = \int_0^{L_B} \lambda(t) \cdot dt = -lnR(L_B) = -ln(1 - F) \cong F(L_B) \tag{7b}$$

Assume that $T_1$ be the time of the first failure, we may express reliability function *R(t)*. That is,

$$R(t) = P(T_1 > t) = P(\text{nofailure in}(0, t]) = \frac{(m)^0 e^{-m}}{0!} = e^{-m} = e^{-\lambda t} \tag{8}$$

If a commercially manufactured product follows the characteristics of the bathtub curve, it might have difficulties achieving success in the field because of the short lifetime and large failure rates early in its life due to design defects. Manufacturers will clearly improve the product design by subjecting its lifetime targets to (1) removing unexpected failures, (2) minimizing random failures for its working period, and (3) extending product lifetime. As the product design improves, the failure rates from the field decease and the product lifetime is extended. For such situations, the established bathtub curve could be converted to a simple line with low failure and longer life that only increases the failure rates toward the termination of its life.

Consequently, because it is less than approximately 20 percent of the cumulative failure rates, the reliability of a mechanical system might be expressed as [27]:

$$R(L_B) = 1 - F(L_B) = e^{-\lambda L_B} \cong 1 - \lambda L_B \tag{9}$$

The reliability of mechanical system therefore might be achieved by estimating the objective product lifetime $L_B$ and failure rate $\lambda$ after optimally identifying the market failure by parametric ALT and modifying the design flaws (or material) of structures (Figure 3).

To achieve the goal of a specific product lifetime by parametric ALT, three cases of the product module were required: (1) a changed module, (2) a newly altered module, and (3) a related module to the former design on the base of field request. The newly designed drawer system in a refrigerator inspected here as a case investigation was a new module which had design faults to be rectified. Consumers were asking for replacements of drawers that had failed early in the life of the refrigerator. The new module B from the market data had a failure rate of 0.24% per year and a B1 life 4.2 years. To address consumer requests, a new lifetime target for the drawer was put to have B1 life 10 years with a cumulative failure rate of one percent (0.1%/year) over the life of the product (Table 1).

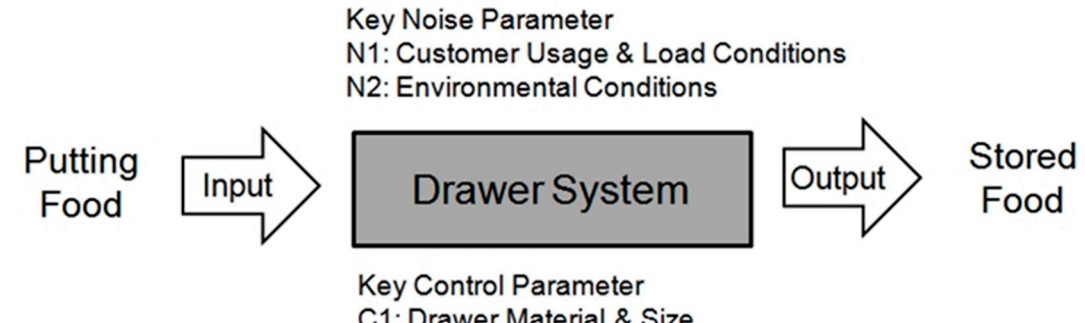

**Figure 3.** Parameter diagram of drawer system.

**Table 1.** Complete ALT scheme of mechanical system such as refrigerator.

| Modules | Field Data | | Expected Reliability | | | | Targeted Reliability | |
|---|---|---|---|---|---|---|---|---|
| | Failure Rate per Year, $\lambda$ (%/Year) | BX Life, $L_B$ (Year) | Failure Rate per Year, $\lambda$ (%/Year) | | | BX Life, $L_B$ (Year) | Failure Rate per Year, $\lambda$ (%/Year) | BX Life, $L_B$ (Year) |
| A | 0.35 | 2.9 | Same | ×1 | 0.35 | 2.9 | 0.10 | 10 (BX = 1.0) |
| B | 0.24 | 4.2 | Newly | ×5 | 1.20 | 0.83 | 0.10 | 10 (BX = 1.0) |
| C | 0.30 | 3.3 | Same | ×1 | 0.30 | 3.33 | 0.10 | 10 (BX = 1.0) |
| D | 0.31 | 3.2 | Altered | ×2 | 0.62 | 1.61 | 0.10 | 10 (BX = 1.0) |
| E | 0.15 | 6.7 | Altered | ×2 | 0.30 | 3.33 | 0.10 | 10 (BX = 1.0) |
| Others | 0.50 | 10.0 | Same | ×1 | 0.50 | 10.0 | 0.50 | 10 (BX = 5.0) |
| Product | 1.9 | 2.9 | - | - | 3.27 | 0.83 | 1.00 | 10 (BX = 10) |

### 2.3. Generalized Time to Failure Model and Sample Size Equation for Parametric ALT

Mechanical systems have subsystems that transmit (generated) power from one component to another through various mechanisms in the system. If there is a design flaw in a component that has an insufficient strength (or stiffness) during repeated loading of the component, it can fail prematurely before achieving its expected lifetime. In reproducing the field failure by parametric ALT, a designer must understand the loading experienced in the field and failure process before being able to both redesign the product shape and change materials selection to achieve the desired reliability of the product. The system must survive the minimum repeated loads in its lifetime so that it can attain the objective reliability (or lifetime) (Figure 4).

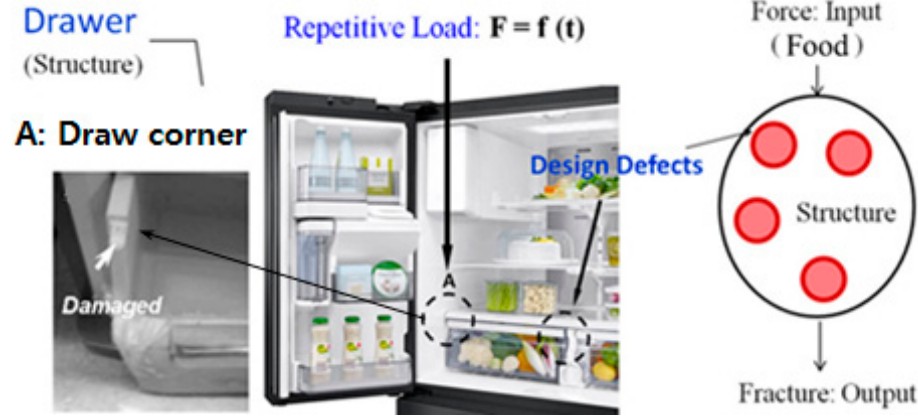

**Figure 4.** Fatigue failure on the product produced by repeated load and design defects.

The primary goal for reliability testing is determining how early the probable failure mode might be obtained in the process. A failure model that connects the relevant coefficients must be derived. That is, the life-stress (LS) model, which includes stresses

and reaction parameters, may clarify numerous failures such as fatigue on the structure. Product failure appears from the emergence of microscopic material imperfections when repetitively subjected to a varying compression and tensile loading under the consumer usage. A life-stress model may be determined from such a viewpoint.

Linear transport processes can be expressed as follow:

$$J = LX \tag{10}$$

where *J* is a flux vector that is associated with some transport property such as mass, momentum, energy, charge, etc. Similarly, *X* is defined as a (driving or thermodynamic) force that is associated with a disequilibrium in some physical property, such as gradients of concentration, fluid velocity, temperature, electrical potential, etc. *L* is a phenomenological transport coefficient, which makes a connection between fluxes and forces.

Transport processes in essence are dissipative when happening in some physical system, which begin to form the system in equilibrium when any net transport comes to stop (Table 2).

**Table 2.** Abridged linear transport phenomena [28,29].

| Ohm's Law of electrical conduction: $j = -\sigma \nabla V$ | | |
|---|---|---|
| *J* = electric current density, *j* (units: A/cm$^2$) | *X* = electric field, $-\nabla V$ (units: V/cm, V = electrical potential) | *L* = conductivity, $\sigma = 1/\rho$ (units: $\rho$ = resistivity ($\Omega$ cm)) |
| Fourier's Law of heat transport: $q = -\kappa \nabla T$ | | |
| *J* = heat flux, *q* (units: W/cm$^2$) | *X* = thermal force, $-\nabla T$ (units: $^\circ$K/cm, T = temperature) | *L* = thermal conductivity, $\kappa$ (units: W/$^\circ$K cm) |
| Fick's Law of diffusion: $F = -D \nabla C$ | | |
| *J* = material flux, *F* (units: /sec cm$^2$) | *X* = diffusion force, $-\nabla C$ (units: /cm$^4$, C = concentration) | *L* = diffusivity, *D* (units: cm$^2$/sec) |
| Newton's Law of viscous fluid flow: $F_u = -\mu \nabla u$ | | |
| *J* = fluid velocity flux, $F_u$ (units: /(sec$^2$ cm)) | *X* = viscous force, $-\nabla u$ (units: /sec, u = fluid velocity) | *L* = viscosity, $\mu$ (units: /(sec cm)) |

For example, when an electric magneto-motive force, $\xi$, is applied, the impurities in materials, generated by electronic motion, easily migrate as the levels of junction energy are lessened. The processes for evaluating solid-state diffusion of impurities in a silicon can be summarized as follows: (1) electro-migration-induced voiding; (2) build-up of chloride ions; (3) trapping of electrons or holes. Solid-state diffusion of impurities in silicon *J* could be stated as [30] (Figure 5):

$$J = [aC(x-a)] \cdot exp\left[-\frac{q}{kT}\left(w - \frac{1}{2}a\xi\right)\right] \cdot v$$

[Density/Area]·[Jump Probability]·[Jump Frequency]

$$= -\left[a^2 v e^{-qw/kT}\right] \cdot cosh\frac{qa\xi}{2kT}\frac{\partial C}{\partial x} + \left[2ave^{-qw/kT}\right]Csinh\frac{qa\xi}{2kT}$$

$$= \Phi(x,t,T)sinh(a\xi)exp\left(-\frac{Q}{kT}\right) \tag{11a}$$

$$= Asinh(a\xi)exp\left(-\frac{Q}{kT}\right)$$

where *A* is constant, *C* is the concentration, *q* is the value of electric charge, $v$ is the frequency, $\Phi()$ is a quantity that does not change its value, *a* is the distance between atoms, $\xi$ is the applied field, *k* is Boltzmann's constant, *Q* is energy, and *T* is temperature.

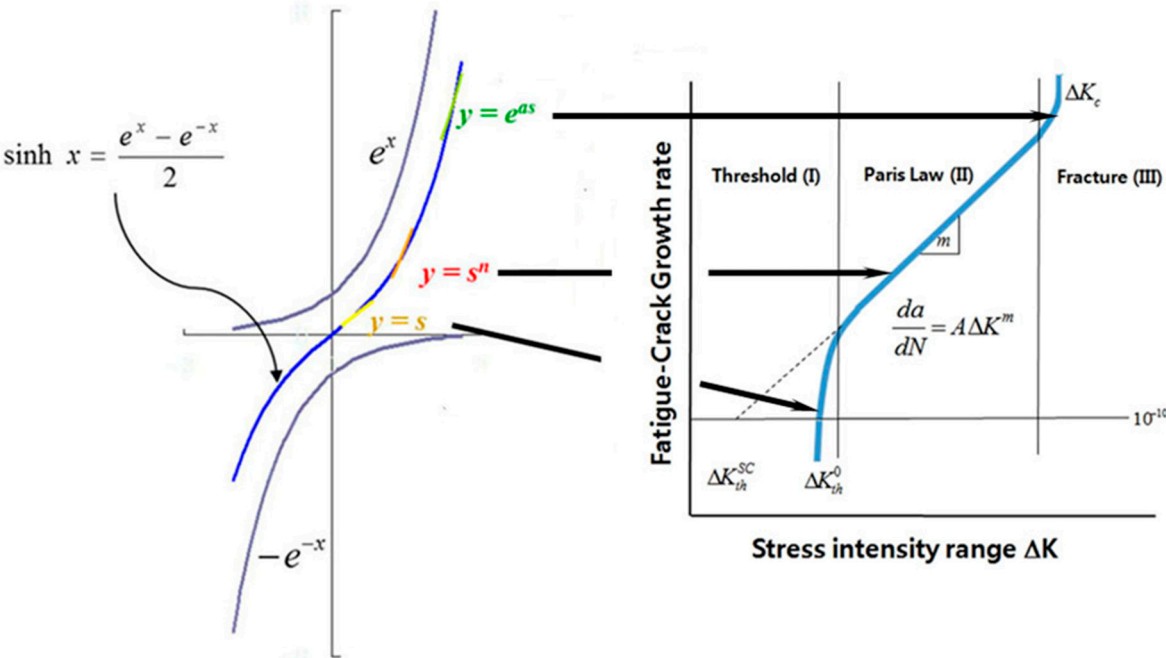

**Figure 5.** Meaning of hyperbolic sine stress term on Paris law.

Conversely, the reaction process that rests on speed might be expressed as.

$$K = K^+ - K^- = a\frac{kT}{h}e^{-\frac{\Delta E - aS}{kT}} - a\frac{kT}{h}e^{-\frac{\Delta E + aS}{kT}} = 2\frac{kT}{h}e^{-\frac{\Delta E}{kT}}sinh(\frac{aS}{kT}) \tag{11b}$$

$$= Bsinh(aS)exp\left(-\frac{\Delta E}{kT}\right)$$

The reaction rate K from Equations (11a) and (11b) might be shortened as

$$K = Bsinh(aS)exp\left(-\frac{Q}{kT}\right) \tag{12}$$

If Equation (12) takes an inverse function, the generalized life-stress (LS) model could be expressed as

$$TF = A[sinh(aS)]^{-1}exp\left(\frac{E_a}{kT}\right) \tag{13}$$

The sine hyperbolic formula $[sinh(aS)]^{-1}$ in Equation (13) may be stated as: (1) $(S)^{-1}$ has a little linear consequence at first, (2) $(S)^{-n}$ has what is considered as medium consequence, and (3) $(e^{aS})^{-1}$ is big in the end (Figure 5).

As ALT is commonly carried out in the medium scope, Equation (13) could be redefined as

$$TF = A(S)^{-n}exp\left(\frac{Q}{kT}\right) \tag{14}$$

where $n = -\left[\frac{\partial ln(TF)}{\partial ln(S)}\right]_T$, $Q = -\left[\frac{\partial ln(TF)}{\partial ln\left(\frac{1}{T}\right)}\right]_S$.

For a specified crack and part geometry, Equation (14) can also be expressed as

$$TF = B(\Delta K)^{-n}exp\left(\frac{Q}{kT}\right) \tag{15}$$

where B is constant, stress intensity factor $\Delta K = YS(or\ \Delta\sigma)\sqrt{\pi a}$,

That is, a (fluctuating) stress range, *S*, moves the crack to develop at some rate. As a certain stress range (or intensity range $\Delta K$) is exerted to a material for some number of cycles $\Delta N$, it moves the crack to develop in length by a particular quantity $\Delta a$. We know that the crack growth rate at a specific stress intensity range is thus obtained by the derivative $\Delta a / \Delta N$ that is dependent on part geometries such as holes, thin area, grooves, etc. So, the crack grows until it gets to a critical size and failure happens. The cause for this acceleration in growth is that the growth rate is dependent on the stress intensity factor at the crack tip, such as holes, notch, etc., and the stress intensity factor is dependent on the crack size, *a*.

The stress of a mechanical system is not a simple quantity to calculate for accelerated testing. If the power is stated as the process of multiplying flows and effort, the stresses originate from effort in a multi-port system [31].

Because the stress due to numerous energy fields in Equation (11a,b) for a mechanical system originates from effort, Equation (14) or (15) could be redefined as

$$TF = A(S)^{-n} exp\left(\frac{E_a}{kT}\right) = C(e)^{-\lambda} exp\left(\frac{E_a}{kT}\right) \tag{16}$$

where *C* is constant.

The acceleration factor (*AF*) could be restated as the proportion between the sufficient elevated stress levels and usual working conditions. From Equation (16), *AF* could be adjusted to integrate the effort notions:

$$AF = \left(\frac{S_1}{S_0}\right)^n \left[\frac{E_a}{k}\left(\frac{1}{T_0} - \frac{1}{T_1}\right)\right] = \left(\frac{e_1}{e_0}\right)^\lambda \left[\frac{E_a}{k}\left(\frac{1}{T_0} - \frac{1}{T_1}\right)\right] \tag{17}$$

To acquire the number of mission cycles of a parametric ALT from the target BX lifetime on the test scheme, the sample size formulation integrated with acceleration factors needs to be determined (See Appendix A).

$$n \geq (r+1) \times \frac{1}{x} \times \left(\frac{L_B^*}{AF \cdot h_a}\right)^\beta + r \tag{18}$$

where the sample size formulation in Equation (18) may be expressed as $n \sim$ (failure numbers + 1)·(1/accumulative failure rate)·((target lifetime/(testing plan time)) ^ $\beta$ + *r*.

Equation (18) also may be confirmed as [32]. That is, for $n \gg r$, sample size formulation may be stated as:

$$n = -\frac{\chi_\alpha^2(2r+2)}{2m^\beta ln R_L} = \frac{\chi_\alpha^2(2r+2)}{2m^\beta ln R_L^{-1}} = \frac{\chi_\alpha^2(2r+2)}{2m^\beta ln(1-F_L)^{-1}} = \frac{\chi_\alpha^2(2r+2)}{2} \times \frac{1}{ln(1-F_L)^{-1}} \times \left(\frac{L_B}{h}\right)^\beta \tag{19}$$

where $m \cong h/L_B$.

Otherwise, for $r = 0$, the sample size formulation may be stated as:

$$n = \frac{ln(1-C)}{m^\beta ln R_L} = \frac{-ln(1-C)}{-m^\beta ln R_L} = \frac{ln(1-C)^{-1}}{m^\beta ln R_L^{-1}} = \frac{ln\alpha^{-1}}{m^\beta ln R_L^{-1}} = \frac{\chi_\alpha^2(2)}{2} \times \frac{1}{ln(1-F_L)^{-1}} \times \left(\frac{L_B}{h}\right)^\beta \tag{20}$$

where $2ln\alpha^{-1} = \chi_\alpha^2(2)$.

So, we know that Equations (19) and (20) have the same formulation as Equation (18).

If the life of a mechanical system, such as a drawer, is targeted to be a B1 life 10 years, the mission cycles might be attained for an assigned collection of samples subjected to repeated (food) loading. In performing parametric ALTs, the design flaws of a newly designed mechanical system could be recognized to help fulfill the lifespan target [33–35].

### 2.4. Case Study—Enhancing the Fatigue Life of a New Drawer System in Domestic Refrigerator

One popular type of refrigerator is the French door refrigerator. Figure 6 shows one with a new designed drawer system. It is made up of a box, two guide rails, and a support in the center between the two drawers. Food is stocked in the drawers. The drawer system should be designed to sustain the working circumstances subjected to it by the consumer. In the United States, the representative customer opens drawers in the refrigerator to store food from five to ten times per day. Stocking food in the French-door refrigerator involves repetition: (1) the drawer is opened, (2) food is placed in it, and then (3) it is closed. The drawers have different amounts of food stored in them when the customer utilizes it.

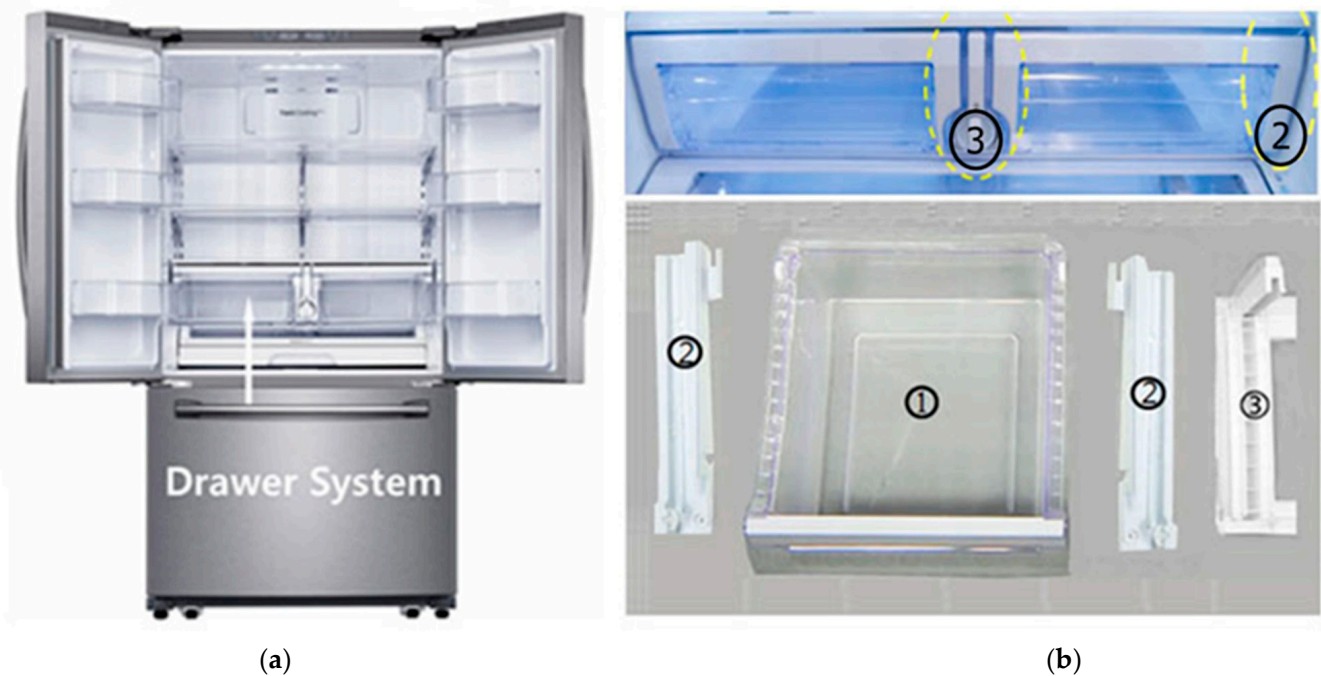

(**a**) (**b**)

**Figure 6.** French-door refrigerator and drawer system. (**a**) French-door refrigerator; (**b**) mechanical components in a drawer: box ①, two guide rails ②, support center ③.

The drawer system returned from the market had been fracturing, causing customers to ask for replacements. As the drawers were failing from being subjected to repeated stresses from openings/closings, it was clear that there was a design problem with the drawer system. Failed drawers from the field showed the drawer system had crucial design defects, which include stress risers—thin ribs and sharp corner angles. These defects created failures (cracks) in the drawers once they were subjected to repeated opening and closing with food loads in the drawers even though the drawer structure was designed to endure repeated food loading under anticipated customer working conditions (Figure 7).

When customers opened the drawer, they usually took food out or put food into it. Relying on the end-user working conditions, the drawer system experienced repeated loading as food was loaded/unloaded and the drawer was opened and shut. To correctly work the drawer system, many mechanical structural parts in drawer assembly should be designed to handle the expected loading from the consumers. Because the concentrated stresses in a mechanical system occur at stress raisers such as sharp corner angles, it is crucial to determine these design flaws experimentally and then modify them.

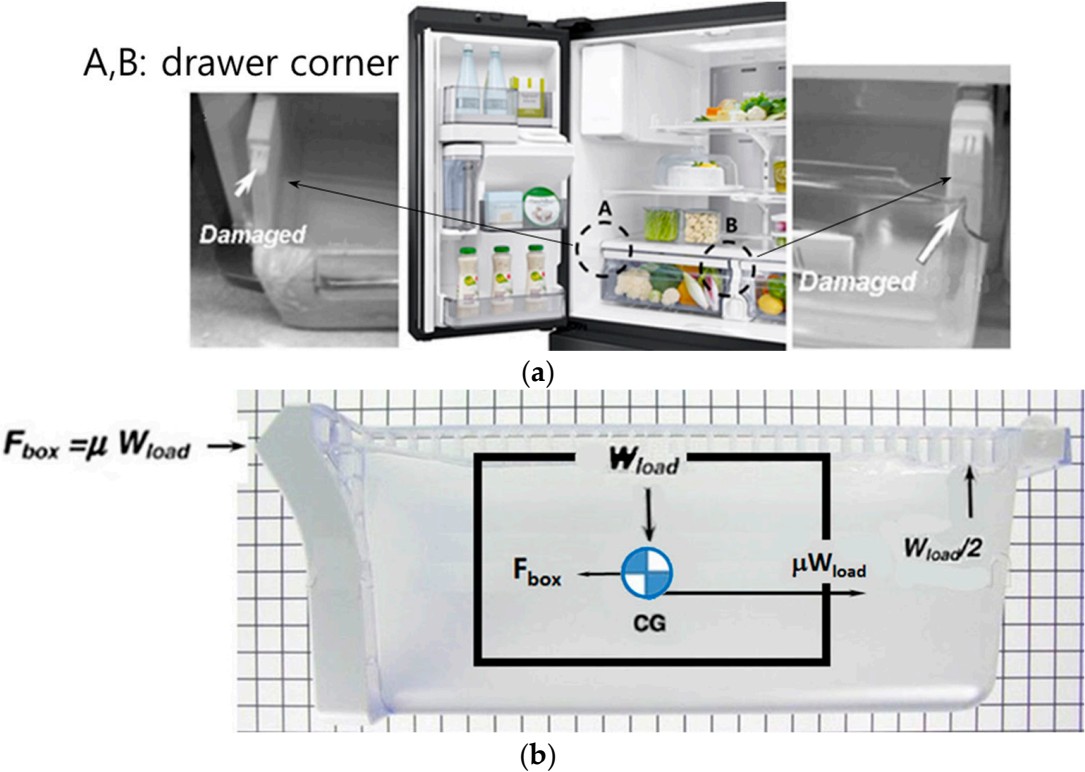

**Figure 7.** Damaged drawer structure after usage. (**a**) damaged drawer in field; (**b**) free-body diagram.

From the drawer system and its free-body diagram (Figure 7b), we knew that the drawing force came from the food weight. The exerted force in the drawer can be expressed as

$$F_{box} = \mu W_{laod} \tag{21}$$

As the stress in a drawer system relies on the exerted food load, Equation (16) could be represented as

$$TF = A(S)^{-n} = A(F_{box})^{-\lambda} = A(\mu W)^{-\lambda} = B(W)^{-\lambda} \tag{22}$$

where $A$ and $B$ are constants

Therefore, the $AF$ in Equation (17) can be restated as follows:

$$AF = \left(\frac{S_1}{S_0}\right)^n = \left(\frac{F_1}{F_0}\right)^{\lambda} = \left(\frac{W_1}{W_0}\right)^{\lambda} \tag{23}$$

For the French-door refrigerator, including the drawer system, the environmental (or working) customer conditions are approximately 0–43 °C with 0.2–0.24 g's of acceleration, and a relative humidity ranging from 0 to 95%. As previously mentioned, the drawer cycles per day were between 5 and 10 times. With the design criterion of a product lifetime for 10 years, $L_B^*$, the drawer structure was subjected to 36,500 usage cycles for the worst occasion.

Under a lifetime target—B1 life 10 years, if the number of lifetime cycles $L_B^*$ and AF are computed for assigned sample size, the actual mission cycles, $h_a$, might be determined from Equation (18). ALT equipment may thus be built and operated in accordance with the expected consumer usage conditions of the drawer system. Through parameter ALTs we might attain the design missing parameters (or design flaws) for a new mechanical system.

The greatest force exerted by the customer in storing food, $W_1$, was 0.059 kN (6 kg$_f$). To decide the stress level for ALT, we used the step-stress life test that can assess the lifetime

under constant used-condition for diverse elevated food weights such as 0.088 kN (9 kg$_f$), 0.117 kN (12 kg$_f$), and 0.137 kN (14 kg$_f$) [36]. As the different stress level is changed, the failure cycles of the drawer system at a specific stress level can be noticed. That is, the crack at first grows slowly, but the growth accelerates (i.e., $da/dN$ increases) as the crack size increases due to accelerated load. So, we can pinpoint the failure time, in which it reaches a critical size and failure happens at the design weak points.

For ALT, the exerted force, W$_2$, took double to 0.117 kN (12 kgf). With a cumulative damage exponent, $\lambda$, of 2, the AF was 4.0 from Equation (23). To acquire the missing design parameters of a new drawer structure, the lifetime target could be put to be more than B1 life 10 years. At first, we supposed that the shape parameter $\beta$ was 2.0, the real test cycles computed from Equation (18) were 37,000 cycles for six sample units. If the parametric ALT fails less than once during 37,000 cycles, the lifetime for drawer structure will be assured to be B1 life 10 years

To signal the number of test cycles, beginning, and ending of the equipment, etc., a testing apparatus with control console was utilized to operate the samples. As the start knob on the controller console gave the starting signal, the straightforward hand-shaped arms pushed and pulled the drawer. The greatest mechanical food force due to accelerated load (0.117 kN) was exerted to the drawer system (Figure 8).

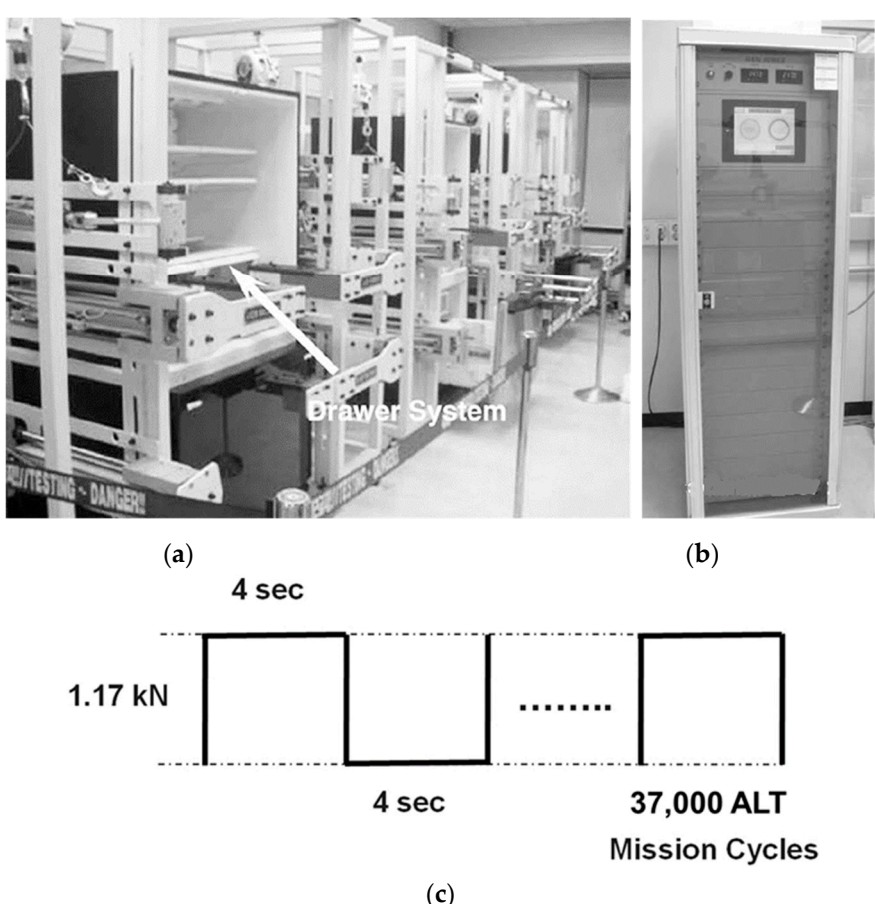

(a)             (b)

(c)

**Figure 8.** ALT apparatus and duty cycles of repeated food load F. (**a**) Apparatus; (**b**) Controller; (**c**) Duty cycles of repeated food load F.

### 3. Results and Discussion

In the 1st ALT, we found the failure time of the following stress levela—0.088 kN (9 kgf), 0.117 kN (12 kgf), and 0.137 kN (14 kgf). For 0.088 kN (9 kgf), the cover of the drawer fractured at 14,000 cycles, 19,000 cycles, and 21,000 cycles. For 0.117 kN (12 kgf), it fractured at 3800 cycles and 4800 cycles. For 0.137 kN (14 kgf), it fractured at 550 cycles,

650 cycles, and 800 cycles. To investigate the fracture surfaces, they were observed by SEM. We found the voids produced because of falling out particles (Figure 9).

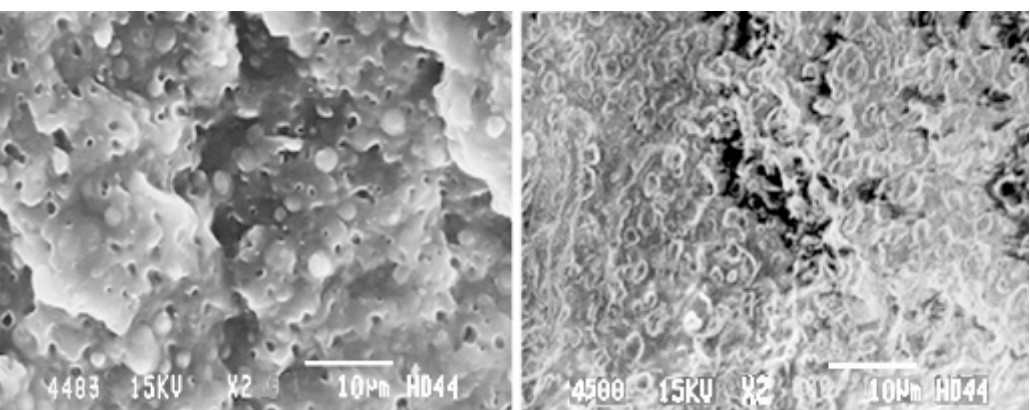

**Figure 9.** SEM micrographs of fracture surfaces.

Finally, we determined the stress level as 0.117 kN (12 kgf) for the parametric ALT because it had a relatively acceptable data—linearity, contrasted with the other stress level. It also can be seen that increasing the repeated food weight has the effect of shifting the left of failure time as stress range (or intensity range $\Delta K$) increased and the crack growth rate moved up, but it did not influence the gradient of the growth rate curve and shape parameter, β (Figure 10).

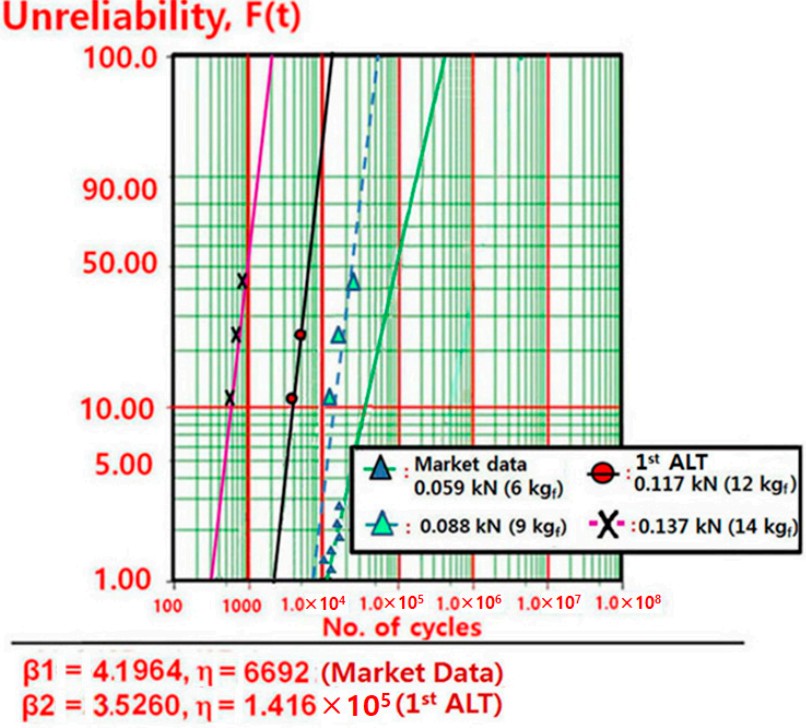

**Figure 10.** Result of the step-stress life test on a Weibull plot.

Initially, when 1.17 kN (12 kgf), as the accelerated weight in drawer was loaded, the left/right rollers on the rail were taken away and the center support rail was deformed so that the drawer system no longer slid. Because of insufficient strength due to design flaws, the draw system was modified by extending the roller support to 7 mm (C2) on the guide rail, as well as adding up strengthened ribs on the center support rail (C1) (Figure 11 and Table 3).

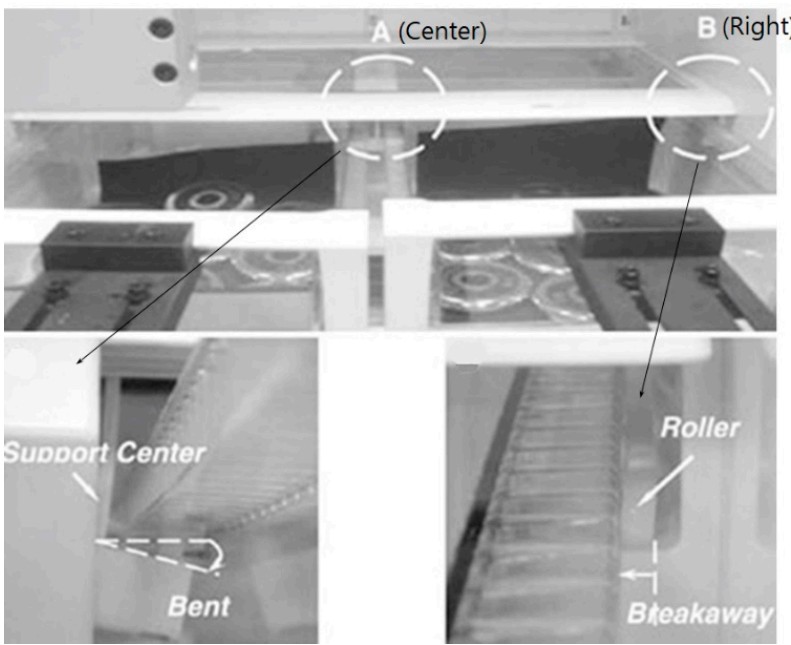

**Figure 11.** Design issues of drawer system in (static) loading.

**Table 3.** Altered designs of center support and rail in initial loading.

| Center Support | Rail |
|---|---|
| 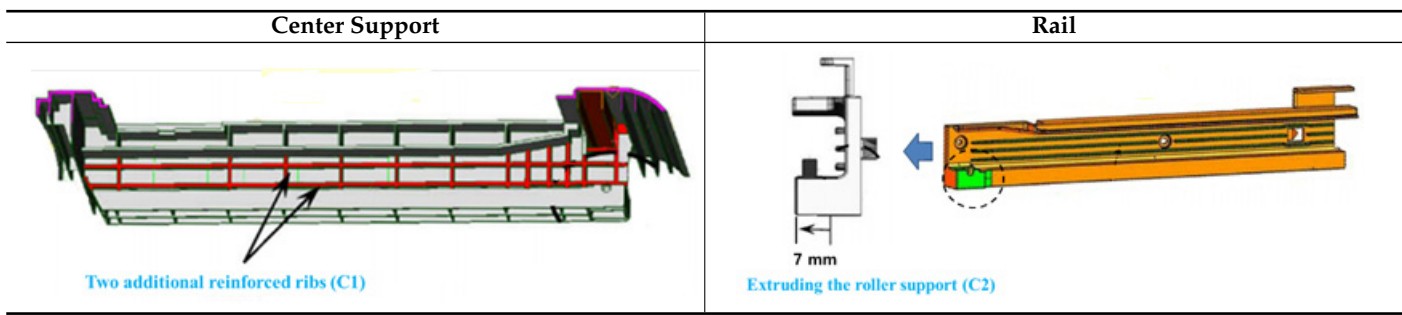 | |

When carefully observing the returned drawers from the marketplace and the first ALT, failure locations were found to be in the junction regions of the drawer cover and its body structure, as a consequence of high repeated stress and expected high da/dN. Figure 10 also provides a Weibull plot of the ALT results, compared to the data from the market. As two patterns had similar slopes on its plot, it was recognized that each loading of the first ALT and the market was similar for the operation conditions. For the shape parameter, $\beta$, the final shape parameter from the chart was affirmed to be 4.2 (Figure 10), compared with the estimated value—2.0. Based on both test consequences and the Weibull plot, the ALT was effective because it identified the design flaws that were accountable for the field failures. Based on the pictures from the ALT and the field and Weibull plot, the tests helped in identifying the problems in the design responsible for market failures and the poor product lifetime.

Due to design defects such as no corners in the high stress areas of intersection (A), the repeated loading of the drawers in conjunction with these structural defects may have been fracturing the drawer cover. These design defects can be altered by: (1) making thicker reinforced ribs, Rib1, C3, from T2.0 mm to T3.0 mm; (2) applying the fillets, Fillet1, C4, from R0.0 mm to R1.0 mm (Figure 12).

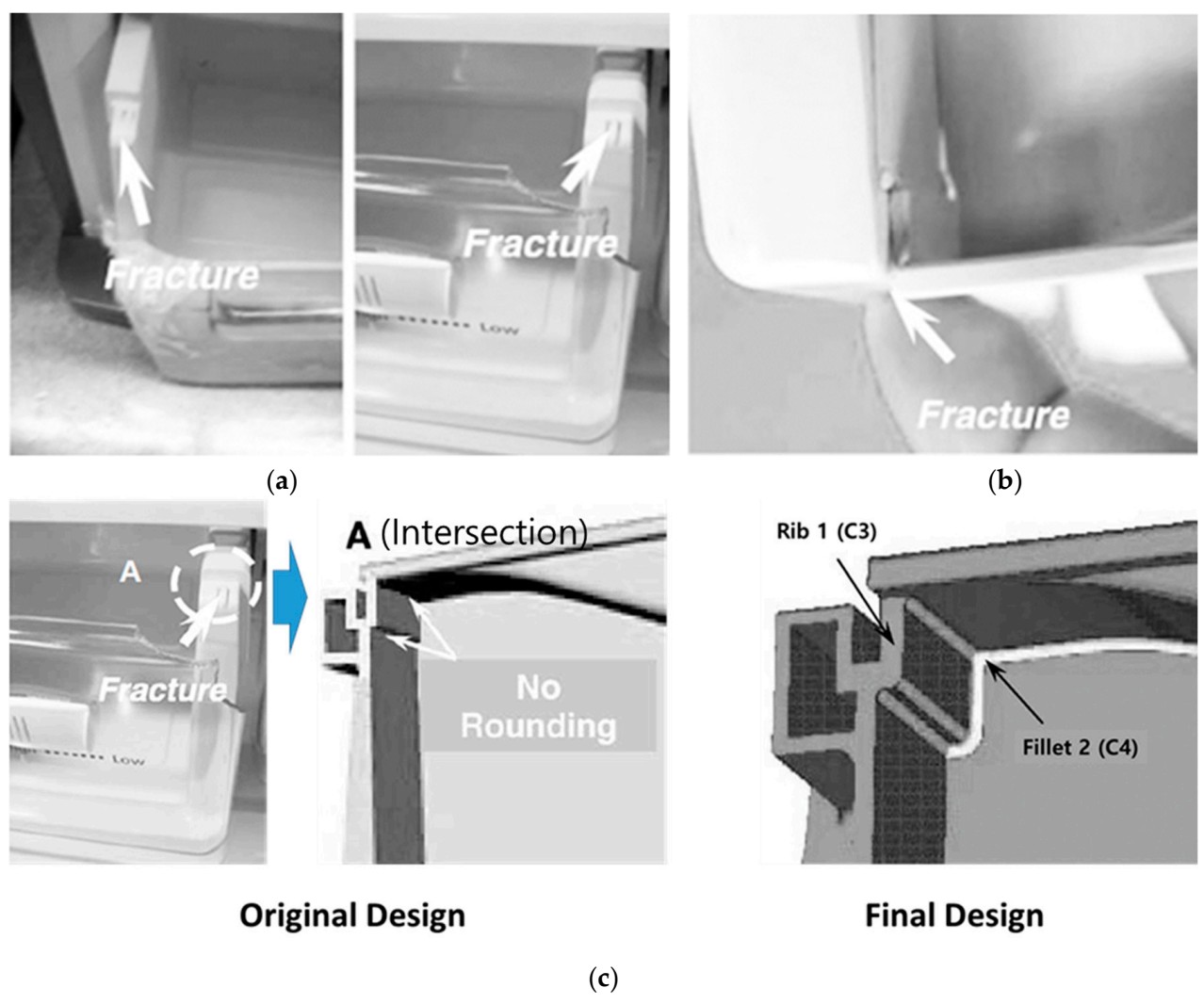

**Figure 12.** Design modification of drawer parts in the first ALT. (**a**) Problematic product after first ALT; (**b**) product with crack in market; (**c**) design modification of drawer parts.

Stress analysis, which may be integrated with fatigue analysis and parametric ALT, was performed by utilizing a finite element analysis (FEA). As the drawer system was attached against the wall, the uncomplicated food loads, as seen in Figure 7, were exerted. Using materials and processing conditions similar to those of the drawer, the constitutive properties of the materials such as polypropylene (drawer structure) were decided. For the old and new designs, we individually evaluated the maximum stresses. Based on this analysis, we could estimate the stress for the new designs for the drawer. After altering the current designs to enhance the fatigue design, the approximated stress concentrations at the intersection areas of the drawer decreased from 23.0 to 14.0 MPa using the FEM analysis. We anticipated that the new design might be successful in lessening fatigue failure of the drawer when subjected to repeated food loading under the customer usage conditions.

With the confirmed shape parameter β being 4.2, the real mission cycles determined from Equation (18) were 19,000 cycles for six samples. If the drawer structure failed at less than once for 13,000 cycles, its lifetime would be assured to be B1 life 10 years. In the second ALT, the fractured guide rail and sunken roller in center support rail failed at 6000 cycles. When carefully inspecting the product failure in the second ALT, the guide rail in drawer structure had no reinforced rib and insufficient corner rounding to endure the repeated loading of the drawer. To improve the guide rail, it was altered by (1) adding reinforced ribs, C5; (2) extending the corner rounding, C6, from R3 mm to R4 mm. The

center support rail in drawer structure was altered by (1) extending the roller rib, C7, from L0.0mm to L2.0mm (Figure 13 and Table 4).

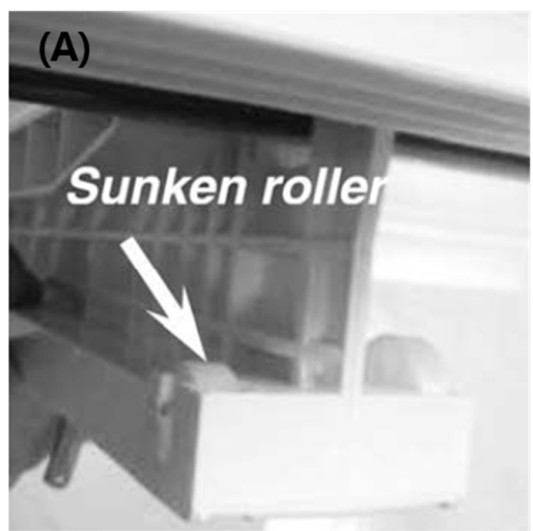
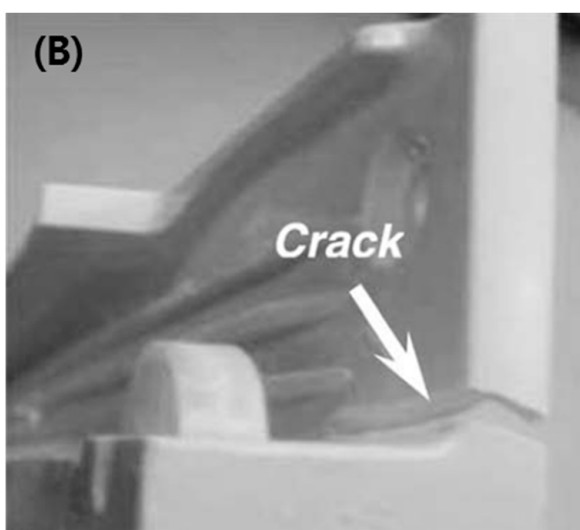

**Figure 13.** Design issues of drawer parts in the second ALT. (**A**) Sunken roller on support rail, (**B**) cracked guide rail.

**Table 4.** Summary for redesigned center support and (left/right) rail.

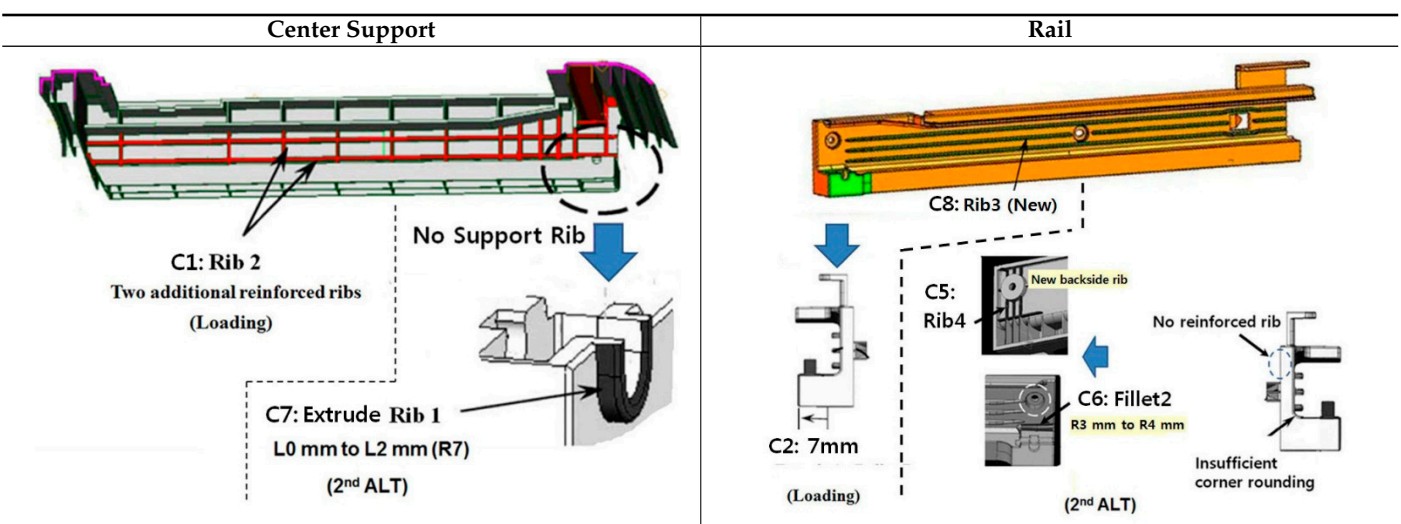

As the drawer structure was upgraded, the design of the new samples was expected to be more than the lifetime target—B1 life 10 years. To confirm the reliability of the design of drawer, a third ALT was performed. Because the confirmed value, β, on the Weibull plot was 4.2, for the lifetime target—B1 life 10 years—the actual mission cycles in Equation (18) were 19,000 for a sample size of six. In the third ALT, there were no issues in the drawer structure until the tests reached 22,000 cycles. Thus, the altered designs acquired from first and second ALT were successful.

Table 4 is a summary for the redesigned center support and (left/right) rail. With the adjusted design parameters, the drawer structure system was assured to reach the lifetime goal—B1 life 10 years.

## 4. Summary and Conclusions

A reliability methodology for application to new mechanical systems was presented. It contained: (1) a parametric ALT scheme created on product BX lifetime, (2) a load inspection for ALT, (3) ALTs with the design alterations, and (4) an estimation of whether the system design(s) fulfilled the target BX lifetime. We thus proposed a generalized stress model and sample size formulation. As an example, a new refrigerator drawer system subjected to repetitive food loading was discussed.

- For the original drawer design, when the accelerated loads were performed, the left/right rail rollers were taken away and the center support rail was deformed because of the insufficient strength. The drawer structure might be altered by extending the roller support in the guide rail as well as attaching up-strengthened ribs on the center support.
- In the first ALT, the box cover in the drawer system fractured at the junction of the drawer cover and its body structure. It was corrected by making the enforced ribs and executing fillets thicker. As the approximated stress concentrations at the intersection areas of drawer decreased from 23.0 to 14.0 MPa using the FEM analysis, we also verified the appropriateness of new design for fatigue.
- In the second ALT, rollers in the center support were broken away and the guide rails fractured. To improve them, the center support rail extended the roller rib. The guide rail also was altered through: (1) adding the reinforced rib; (2) enhancing the corner rounding.
- With these altered designs, in the third ALT, there were no reliability issues. The altered drawer system was guaranteed to satisfy the lifetime need—B1 life 10 years. By examining problematic field products, load analysis, and parametric ALTs with design modifications, the mechanical system, such as in the drawer, was successful in having a lengthy lifetime.
- By figuring out the design problems for returned products, we could design and perform parametric ALTs. After reproducing the design failures, we might modify them. Finally, we checked if the product achieved the lifetime target. In the process, we utilized the (generalized) stress model and sample size equation.

This structured reliability method can be applied to other mechanical products such as airplanes, automobiles, and construction machinery. To make use of this systematic method, designers might recognize why products fail during their lifetime. If there are design flaws in the structure which are subjected to repeated loads, the system will be unsuccessful before its anticipated lifetime. After identifying the load features of a mechanical system, engineers may perform the parametric ALT until the needed mission cycles. Ultimately, parametric ALT can be utilized to recognize the design problems of a mechanical product and alter them.

**Author Contributions:** S.W. performed the idea developing, methodology, analysis and experiment, and wrote down the original document. D.L.O. inspected the study and writing for the draft manuscript. Y.M.H. corrected the document. All authors have understood and agreed to the published edition of the document. All authors have read and agreed to the published version of the manuscript.

**Funding:** This research received no external funding.

**Institutional Review Board Statement:** Not applicable.

**Informed Consent Statement:** Not applicable.

**Data Availability Statement:** The data supplied in this research can be attained on request from the corresponding author.

**Conflicts of Interest:** The authors declare no conflict of interest.

## Abbreviations

| $a$ | Crack size |
|---|---|
| $BX$ | Time which is a cumulated failure rate of X%: durability index |
| $E_a$ | Activation energy, eV |
| $e$ | Effort |
| $f$ | Flow |
| $F(t)$ | Unreliability |
| $h$ | Testing cycles (or cycles) |
| $h*$ | Non-dimensional testing cycles, $h^* = h/L_B \geq 1$ |
| $k$ | Boltzmann's constant, $8.62 \times 10^{-5}$ eV/deg |
| $k_a$ | Constant of the counter-electromotive force |
| $L_B$ | Target BX life and x = 0.01X, on the condition that x $\leq$ 0.2 |
| $n$ | Number of test samples |
| $R$ | Ratio for minimum stress to maximum stress in stress cycle, $\sigma_{min}/\sigma_{max}$ |
| $r$ | Failed numbers |
| $R_a$ | Electromagnetic resistance |
| $S$ | Stress |
| $T$ | Temperature, K |
| $t_i$ | Test time for each sample |
| TF | Time to failure |
| $T_L$ | Ice-crushing torque in bucket, kN cm |
| $X$ | Accumulated failure rate, % |
| $x$ | x = 0.01X, on condition that x $\leq$ 0.2. |
| Greek symbols | |
| $\eta$ | Characteristic life |
| $\lambda$ | Cumulative damage exponent in Palmgren–Miner's rule |
| $\chi^2$ | Chi-square distribution |
| $\alpha$ | Confidence level |
| $\mu$ | Friction coefficient |
| Superscripts | |
| $\beta$ | Shape parameter in Weibull distribution |
| $n$ | Stress dependence, $n = -\left[\frac{\partial ln(T_f)}{\partial ln(S)}\right]_T$ |
| Subscripts | |
| 0 | Normal stress conditions |
| 1 | Accelerated stress conditions |

## Appendix A. Derivation of Sample Size Equation for Redesign of Mechanical Systems through ALT

To acquire the number of mission cycles of a parametric ALT from the target BX lifetime on the test scheme, the sample size formulation integrated with acceleration factors needs to be determined.

Currently, numerous methods have been suggested to settle matters on sample size. The Weibayes model, utilizing Weibull examination, is a popularly accepted procedure of inspecting reliability data. However, it cannot be straightforwardly utilized due to the complication of the mathematical equation. The cases as failures ($r \geq 1$) and no failures ($r = 0$) need to be separate. As a consequence, it is practical to obtain a probable sample size formulation that could provide the mission cycle after right presumptions.

The Weibull distribution is popularly utilized because it can be simply described as a shape parameter and characteristic life. That is, if the product pursues Weibull distribution, the accumulated failure rate, $F(t)$, in Equation (2) is stated as

$$F(t) = 1 - e^{-\left(\frac{t}{\eta}\right)^{\beta}} \tag{A1}$$

where $t$ is time, $\eta$ is characteristic life, and $\beta$ is shape parameter.

When $t = L_B$ in Equation (A1), the correlation between BX life, $L_B$, and characteristic life, $\eta$, may be stated as

$$L_B^\beta = \left( ln\frac{1}{1-x} \right) \cdot \eta^\beta \tag{A2}$$

where $x = 0.01F(t)$.

Product failures in the bathtub curve might be categorized into three sections: infant mortality ($\beta < 1$), random failure ($\beta = 1$), and wear-out failure (($\beta > 1$) that may be defined as shape parameter in the Weibull distribution. For approximating lifetime, the Weibayes method is explained as Weibull analysis with an assigned shape parameter that can be presumed from prior learning or actual testing.

For a Weibayes analysis, the characteristic life can be obtained from utilizing the maximum likelihood estimate (*MLE*) as follows:

$$\eta_{MLE}^\beta = \frac{\sum_{i=1}^n t_i^\beta}{r} \tag{A3}$$

where $\eta_{MLE}$ is the *MLE* of the characteristic life, $n$ is the whole number of samples, $t_i$ is the full test length for each sample, and $r$ is the number of failures.

When the failure number is $r \geq 1$ and confidence level is $100(1-\alpha)$, the characteristic life, $\eta_\alpha$, could be approximated as the following middle term and altered into the last term utilizing Equation (A3):

$$\eta_\alpha^\beta = \frac{2r}{\chi_\alpha^2(2r+2)} \cdot \eta_{MLE}^\beta = \frac{2}{\chi_\alpha^2(2r+2)} \cdot \sum_{i=1}^n t_i^\beta \text{ for } r \geq 1 \tag{A4}$$

where there are no failures, we know that $2ln(\alpha^{-1}) = \chi_2^2(2)$. That is, at this moment, the first term, $ln\frac{1}{\alpha}$, is mathematically identical to the Chi-squared value, $\frac{\chi_\alpha^2(2)}{2}$, if $p$-value is $\alpha$. The characteristic life, $\eta_\alpha$, could be defined as follows:

$$\eta_\alpha^\beta = \frac{2}{\chi_\alpha^2(2)} \cdot \sum_{i=1}^n t_i^\beta = \frac{1}{ln\frac{1}{\alpha}} \cdot \sum_{i=1}^n t_i^\beta \text{ for } r = 0 \tag{A5}$$

Thus, Equation (A4) can be applicable to all occasions. In other words,

$$\eta_\alpha^\beta = \frac{2}{\chi_\alpha^2(2r+2)} \cdot \sum_{i=1}^n t_i^\beta \text{ for } r \geq 0 \tag{A6}$$

If Equation (A6) is inserted into Equation (A2), BX life may be expressed as

$$L_B^\beta = \left( ln\frac{1}{1-x} \right) \cdot \eta^\beta \cong x \cdot \eta^\beta = x \cdot \frac{2}{\chi_\alpha^2(2r+2)} \sum_{i=1}^n t_i^\beta \text{ for } x \leq 0.2 \ (20\%) \tag{A7}$$

If the sample size is big enough, the planned testing time might advance as

$$\sum_{i=1}^n t_i^\beta \cong n \cdot h^\beta \tag{A8}$$

where $h$ is the planned test time

The approximated lifetime ($L_B$) in the test might be as lengthy as the targeted life ($L_B^*$). That is, if Equation (A8) is substituted in Equation (A7), we can obtain as follows:

$$L_B^\beta = x \cdot \frac{2}{\chi_\alpha^2(2r+2)} \sum_{i=1}^n t_i^\beta \cong x \cdot \frac{2}{\chi_\alpha^2(2r+2)} \cdot n \cdot h^\beta \geq L_B^{*\beta} \tag{A9}$$

If Equation (A9) is altered, the sample size formulation is redefined as follows:

$$n \geq \frac{\chi_\alpha^2(2r+2)}{2} \times \frac{1}{x} \times \left( \frac{L_B^*}{h} \right)^\beta \tag{A10}$$

For a 60% confidence level, the first expression $\frac{\chi_\alpha^2(2r+2)}{2}$ may be estimated as $(r+1)$. Thus, Equation (A10) may be estimated as follows:

$$n \geq (r+1) \times \frac{1}{x} \times \left(\frac{L_B^*}{h}\right)^\beta \tag{A11}$$

If the acceleration factors in Equation (17) are attached to the planned testing time *h*, Equation (A11) is defined as follows:

$$n \geq (r+1) \times \frac{1}{x} \times \left(\frac{L_B^*}{AF \cdot h_a}\right)^\beta + r \tag{A12}$$

where the sample size formulation in Equation (A12) may be expressed as *n* ~ (failure numbers + 1)·(1/accumulative failure rate)·((target lifetime/(testing plan time)) ˆ *β* + *r*.

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
