# Peer review of "Process to Establish the Enhance of Fatigue Life of New Mechanical System Such as a Drawer by Accelerated Tests"

_applsci, doi:10.3390/app12094497_

Round 1

Reviewer 1 Report

Developing the reliability analysis on other benchmarks and deepening the qualitative analysis performed. The paper can be accepted for publication

Author Response

Dear Sirs:

Thank you for your review of our paper.  Below is a response to the review.

Comment 1. Developing the reliability analysis on other benchmarks and deepening the qualitative analysis performed. The paper can be accepted for publication.

Response:   1) Based on your recommendation, we added the result of finite element analysis and the summary of this systematic reliability method in the conclusion.

2) To simplify and show the methodology, we combined section 2.3 & 2.4 into section 2.3 and retitled Generalized time to failure model and sample size equation for parametric ALT. And we moved section 2.4 to appendix A.

3) And we add the results of finite element analysis and the fracture surfaces observed by SEM as a result of step-stress (Figure 9) in section 3 (results and discussion). Please check it.

Sincerely,

The authors

Reviewer 2 Report

In this paper, the authors present the results of the parametric ALT study as a systematic approach that can be used to identify structural defects of the newly designed mechanical structure and their modification. The work presents the methodology of work reliability for use in new mechanical systems. 

Author Response

Dear Sirs:

Thank you for your review of our paper.  Below is a response to the review.

Comment 1. In this paper, the authors present the results of the parametric ALT study as a systematic approach that can be used to identify structural defects of the newly designed mechanical structure and their modification. The work presents the methodology of work reliability for use in new mechanical systems.

Response:   1) Based on your recommendation, we added the result of finite element analysis and the summary of this systematic reliability method in the conclusion.

2) To simplify and show the methodology, we combined section 2.3 & 2.4 into section 2.3 and retitled Generalized time to failure model and sample size equation for parametric ALT. And we moved section 2.4 to appendix A.

3) And we add the results of finite element analysis and the fracture surfaces observed by SEM as a result of step-stress (Figure 9) in section 3 (results and discussion). Please check it.

Sincerely,

The authors

Reviewer 3 Report

Paper regards accelerated life testing of refrigerator drawer. Testing program is very simple (4 tests with increased service loads), and conclusions regard mostly design modifications. In contrast to this extensive theoretical framework regarding applied reliability methodology is given in this paper. Some parts of this  framework are hardly understandable (“Generalized time to failure model for ALT”) or rather I would say bizarre and have nothing in common with performed simple testing program. Also well-known fracture mechanics approach and stress-life approach for component life estimation are mentioned, but no crack propagation analysis or  stress analysis can be found in this paper. For the clarity of the paper I would suggest authors to focus on obtained experimental results and provide only necessary reliability methodology theoretical framework, which as the matter of fact is based on observations of component failure during service enriched by statistical approach. 

Author Response

Dear Sirs:

Thank you for your review of our paper. This study suggests parametric accelerated life testing (ALT) as a reliability methodology to generate the reliability quantitative (RQ) specifications – mission cycle due to BX lifetime – for verifying a newly designed mechanical product. Thus, it will enable mechanical systems fatigued by repeated impact loads to improve their lifetime. It consists of (1) parametric ALT plan based on product BX lifetime that will be X percent of the accumulated failure, (2) load analysis for ALT, (3) a tailored series of parametric ALTs with the design modifications, and (4) evaluation of whether the product design(s) achieves the target BX lifetime.

Below is a response to the review.

Comment 1. Paper regards accelerated life testing of refrigerator drawer. Testing program is very simple (4 tests with increased service loads), and conclusions regard mostly design modifications.

Response:   Based on your recommendation, we added the result of finite element analysis and the summary of this systematic reliability method. See the conclusion.

Comment 2. In contrast to this extensive theoretical framework regarding applied reliability methodology is given in this paper. Some parts of this framework are hardly understandable (“Generalized time to failure model for ALT”) or rather I would say bizarre and have nothing in common with performed simple testing program.

Response: To simplify and show the methodology, we combined section 2.3 & 2.4 into section 2.3, retitled Generalized time to failure model and sample size equation for parametric ALT, and modified the contents. For section 2.4 (Derivation of sample size equation for redesign of Mechanical Systems through ALT), we moved to appendix A. See the section 2 and appendix A.

Comment 3. Also, well-known fracture mechanics approach and stress-life approach for component life estimation are mentioned, but no crack propagation analysis or stress analysis can be found in this paper. For the clarity of the paper I would suggest authors to focus on obtained experimental results and provide only necessary reliability methodology theoretical framework, which as the matter of fact is based on observations of component failure during service enriched by statistical approach.

Response: As recommended, we add the results of finite element analysis and the fracture surfaces observed by SEM as a result of step-stress (Figure 9) in section 3 (results and discussion). Please check it.

Sincerely,

The authors
